# Regulation of Ebola GP conformation and membrane binding by the chemical environment of the late endosome

Aastha Jain[1], Ramesh Govindan[1,2], Alex R. Berkman[1], Jeremy Luban[3,4], Marco A. Díaz-Salinas[1], Natasha D. Durham[1], James B. Munro[1,4] *

1 Department of Microbiology and Physiological Systems, UMass Chan Medical School, Worcester, Massachusetts, United States of America, 2 Medical Scientist Training Program, Tufts University School of Medicine, Boston, Massachusetts, United States of America, 3 Program in Molecular Medicine, UMass Chan Medical School, Worcester, Massachusetts, United States of America, 4 Department of Biochemistry and Molecular Biotechnology, UMass Chan Medical School, Worcester, Massachusetts, United States of America

* james.munro@umassmed.edu

**Data Availability Statement:** All relevant data are within the manuscript and its Supporting Information files.

## Abstract

Interaction between the Ebola virus envelope glycoprotein (GP) and the endosomal membrane is an essential step during virus entry into the cell. Acidic pH and $Ca^{2+}$ have been implicated in mediating the GP-membrane interaction. However, the molecular mechanism by which these environmental factors regulate the conformational changes that enable engagement of GP with the target membrane is unknown. Here, we apply fluorescence correlation spectroscopy (FCS) and single-molecule Förster resonance energy transfer (smFRET) imaging to elucidate how the acidic pH, $Ca^{2+}$ and anionic phospholipids in the late endosome promote GP-membrane interaction, thereby facilitating virus entry. We find that bis(monoacylglycero)phosphate (BMP), which is specific to the late endosome, is especially critical in determining the $Ca^{2+}$-dependence of the GP-membrane interaction. Molecular dynamics (MD) simulations suggested residues in GP that sense pH and induce conformational changes that make the fusion loop available for insertion into the membrane. We similarly confirm residues in the fusion loop that mediate GP's interaction with $Ca^{2+}$, which likely promotes local conformational changes in the fusion loop and mediates electrostatic interactions with the anionic phospholipids. Collectively, our results provide a mechanistic understanding of how the environment of the late endosome regulates the timing and efficiency of virus entry.

## Author summary

Ebola virus causes disease in humans with high fatality. A better understanding of how Ebola virus enters cells is critical to inform the development of novel therapeutic and preventative measures. The viral glycoprotein present on the surface of the virus mediates attachment to cells and subsequent entry through a poorly understood mechanism involving fusion of viral and cellular membranes. Here, we employ computational and experimental biophysical techniques to understand how the Ebola glycoprotein senses chemical

**Funding:** This work was supported by NIH grants R01AI174645 and R01GM143773 to J.B.M, and R01AI148784 to J.L. The funders had no role in study design, data collection and analysis, decision to publish, or preparation of the manuscript.

**Competing interests:** The authors have declared that no competing interests exist.

cues in its environment, such as pH, calcium ions, and specific lipid species to ensure that entry occurs at the right time and place. Our results specify elements of the glycoprotein that control its structure under changing physiological environments.

## Introduction

Ebola virus (EBOV) causes disease in humans with an average case fatality rate of 50% [1]. Incidences of repetitive outbreaks, changes in virulence, or emergence of resistance could reduce the effectiveness of currently approved vaccination and treatment regimes. Therefore, a better understanding of the EBOV entry mechanism, an under-utilized drug target, would aid in the development of effective prophylactic and treatment interventions. The EBOV envelope glycoprotein (GP), present on the surface of the virion, mediates virus entry into host cells. GP is a trimer of heterodimers with each protomer consisting of two subunits, GP1 and GP2, which are linked by disulfide bonds [2]. Virions attach to the host cell surface via GP1 interaction with C-type lectins and phosphatidylserine receptors, and are internalized by macropinocytosis [3,4]. Once inside the endosomes, cathepsins B and L proteolytically remove the mucin-like domain and glycan cap from GP1, enabling its binding to the endosomal receptor, the Niemann-Pick C1 (NPC1) cholesterol transporter [5–8]. Conformational changes in GP, which occur in response to cues encountered in the endosome, reposition the fusion loop of GP2 from a hydrophobic cleft in the neighboring protomer to a position where it can insert into the endosomal membrane. Additional factors yet to be identified are likely required to trigger the complete cascade of conformational changes in GP needed to promote fusion of the viral and endosomal membranes [9].

While in the endocytic pathway, acidification of the endosomal lumen and $Ca^{2+}$ play critical, but poorly defined roles in promoting EBOV entry [10–14]. Our previous single-molecule Förster resonance energy transfer (smFRET) experiments using intact, trimeric GP on the surface of pseudoviral particles demonstrated that low pH and $Ca^{2+}$ promote conformational changes in GP that correlate with lipid mixing [12]. A recent study further showed that in addition to low pH and $Ca^{2+}$, cathepsins enhanced the fusion of GP-containing pseudoparticles with supported phospholipid bilayers formed from endosomal membranes [15]. In a prior study of a GP fragment (residues 507–560) containing the fusion loop, conserved residues were mutated to identify pH sensors that control membrane binding and fusion [16]. Only mutation of H516 reduced lipid mixing by 80% in an *in vitro* assay. However, no single amino acid within the fusion loop could be conclusively identified as a pH sensor that could trigger efficient lipid mixing or cell entry. Another study investigated the interaction of $Ca^{2+}$ ions with anionic residues flanking the fusion loop [14]. Residues D522 and E540 were crucial for interaction with $Ca^{2+}$ of a GP fragment containing the fusion loop, which enhanced membrane binding. While these studies suggested residues in the fusion loop that are involved in sensing pH and $Ca^{2+}$, the role of these residues in mediating conformational changes of trimeric GP has not been evaluated. Nor has it been determined how additional residues outside of the fusion loop might allosterically regulate conformational dynamics of GP and fusion loop-mediated membrane binding.

Lipids in host cell membranes can facilitate virus attachment and regulate fusion. The lipid content of endosomes is important in sorting of enveloped viruses into specific compartments and avoiding premature fusion [17]. The late endosomal membrane is rich in the anionic lipids phosphatidylserine (PS) and bis(monoacylglycero)phosphate (BMP) [18,19]. BMP is essential for fusion of viruses such as Dengue, Lassa, Uukuniemi and vesicular stomatitis virus that

enter cells through the endocytic route [20–24]. However, a role for BMP in EBOV GP entry could not be verified using a cell-cell fusion assay [9]. This may indicate that the plasma membrane lacks other lipid or protein components, which are specific to the endosome and essential for EBOV GP-mediated fusion. Therefore, whether endosomal lipids play a role in EBOV entry remains an open question.

In the present study, we sought to elucidate the mechanistic basis for how the chemical features of the late endosome enable GP to engage the target membrane prior to fusion. To probe the ability of trimeric GP to bind membranes of defined composition, we developed a fluorescence correlation spectroscopy (FCS) assay. FCS provides quantitative information on the diffusion of molecules in solution. In comparison to conventional pull-down or membrane flotation methods, FCS has high spatio-temporal resolution, requires low sample quantity, and provides rapid experimental throughput [25]. Our results indicate that the anionic lipids PS and BMP enhance the GP-membrane interaction. Furthermore, we demonstrate that BMP is the primary mediator of $Ca^{2+}$-dependence for this interaction. Parallel smFRET imaging indicated that acidic pH is the primary driver of global GP conformational changes, which move the fusion loop away from its hydrophobic cleft to a position where it can engage the membrane. $Ca^{2+}$ binding to the fusion loop may have a more localized effect on GP conformation. The results of mutagenesis support a model in which pH- and $Ca^{2+}$-sensing residues tune the responsiveness of GP to the chemical environment of the late endosome, ensuring proper timing of conformational changes necessary for fusion.

## Results

### Anionic lipids and $Ca^{2+}$ mediate GP-membrane binding at acidic pH

We first developed an *in-vitro* FCS assay to quantify the interaction between GP and model membranes (**Fig 1A**). The trimeric ectodomain of GP (GPΔTM) from the Mayinga strain of EBOV, lacking the transmembrane and mucin-like domains, was expressed and purified in Expi293 cells. GPΔTM was site-specifically labelled with Cy5 using an enzymatic labelling approach (Materials and Methods) [26]. After proteolytic removal of the glycan cap with HRV3C from labelled GPΔTM (forming GP^CL), FCS measurements showed that GP^CL diffused as a single homogeneous species with a diffusion time of 0.308 ± 0.007 ms (**Fig 1B**). We then prepared 100-nm liposomes using phosphatidylcholine (PC), cholesterol (Ch), and the lipophilic fluorophore DiD (PC:Ch:DiD 95:5:0.04mol%). FCS indicated a diffusion time of 2.6 ± 0.2 ms for the liposomes, reflecting their larger size and slower motility than GP^CL (**Fig 1C**). To facilitate GP^CL-membrane interactions under conditions that approximate the late endosome, we incubated GP^CL with unlabeled liposomes (PC:Ch 95:5mol%) at 37˚C for 20 mins at pH 5.5. Evaluation with FCS indicated a mixture of two species with diffusion times consistent with unbound GP^CL (*ca*. 0.3 ms), and GP^CL bound to liposomes (*ca*. 2.6 ms). Under these conditions, 4.8 ± 0.1% of GP^CL bound to the zwitterionic liposomes (**Fig 1D**). Introduction of $Ca^{2+}$ at concentrations as high as 1 mM had minimal effect.

We then sought to optimize the lipid composition such that we could robustly detect GP^CL binding to liposomes and later evaluate the role of specific GP^CL residues and $Ca^{2+}$. We specifically asked whether inclusion of anionic phospholipids found in the late endosome, such as PS and BMP, might further promote GP^CL-membrane interaction. Introduction of PS into the liposomes (PC:PS:Ch 55:40:5mol%) increased GP^CL binding to 10.9 ± 0.2% in the absence of $Ca^{2+}$. Addition of $Ca^{2+}$ at concentrations of 1 μM and 10 μM further increased GP^CL binding to 17.9 ± 0.3% and 27.5 ± 0.2%, respectively (**Fig 1D**). Further increases in $Ca^{2+}$ concentration reduced GP^CL binding. Inclusion of BMP (PC:PS:BMP:Ch 40:40:15:5mol%), an anionic lipid specific to late endosomes [19,27], further increased GP^CL-membrane interaction to

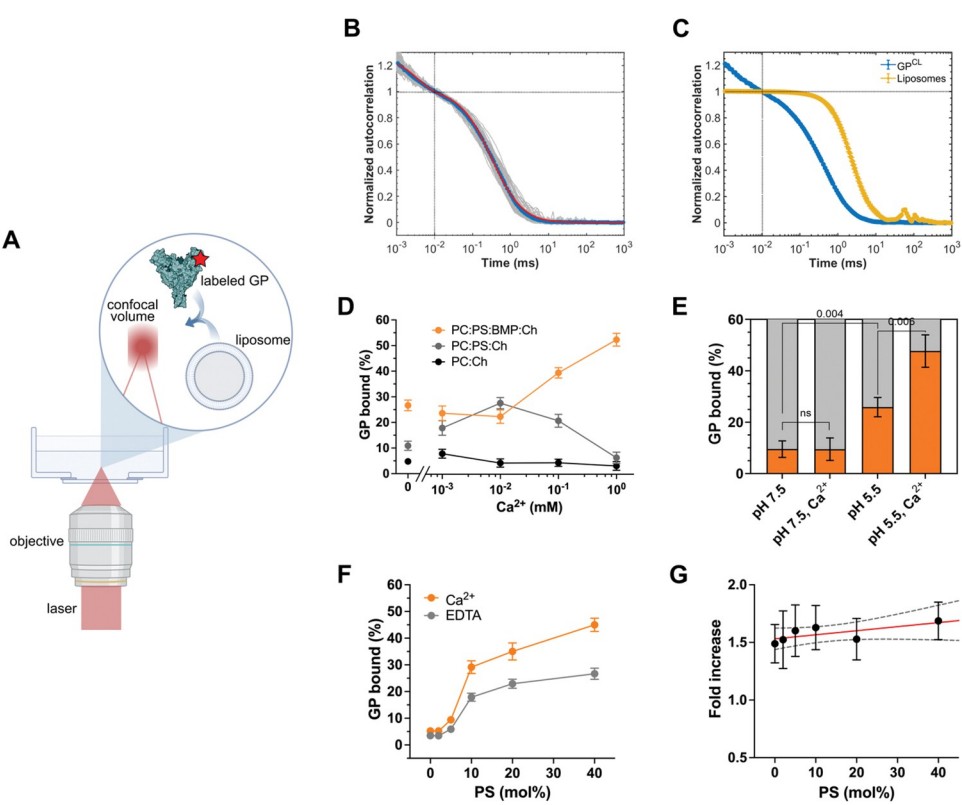

**Fig 1. Anionic phospholipids and Ca²⁺ promote GP^{CL}-membrane interaction.** (**A**) Experimental setup of FCS assay for quantifying GP-membrane interactions. A confocal spot is positioned in a dilute solution of Cy5-labelled GP^{CL} with or without liposomes. Fluorescent particles are detected upon diffusion through the confocal volume. (**B**) Autocorrelation data obtained from unbound Cy5-labelled GP^{CL}. The autocorrelation of individual 5-sec fluorescence intensity traces, normalized to the point at $10^{-2}$ ms, was calculated (grey curves). The average of 100 autocorrelation curves (blue) was fitted to a model for a single fluorescent species diffusing in 3 dimensions with photophysical (triplet state) dynamics (red; see Materials and Methods). This analysis indicated that GP^{CL} has a diffusion time of $\tau_{protein} = 0.308 \pm 0.007$ ms. Photophysical dynamics were observed below $10^{-2}$ ms. (**C**) Autocorrelation curves comparing diffusion times of GP^{CL} (blue; determined in (B)) and liposomes (yellow; determined in the same manner). The diffusion time of liposomes is $\tau_{liposome} = 2.6 \pm 0.2$ ms, which is 8-fold longer than GP^{CL}, allowing for identification of membrane-bound and unbound GP^{CL} in solution. (**D**) The percentage of GP^{CL} bound to unlabelled liposomes determined by fitting autocorrelation data to a model for two fluorescent species diffusing in 3 dimensions, liposome-bound and unbound GP^{CL} with diffusion times of 2.6 ms and 0.3 ms, respectively (Materials and Methods). GP^{CL} binding to liposomes of varying lipid compositions was measured at pH 5.5 as a function of Ca²⁺ concentration: PC:Ch (black circles), PC:PS:Ch (grey circles) and PC:PS:BMP:Ch (orange circles). (**E**) Percentage of total GP^{CL} bound to PC: PS:BMP:Ch liposomes under the indicated conditions. To ensure sensitivity to changes in the percentage of membrane binding of GP^{CL} variants, a total lipid concentration of 0.5 mM was used such that a maximum of 50% wild-type GP^{CL} bound to liposomes at low pH in the presence of Ca²⁺. Indicated $p$-values were determined by $t$-test (ns, $p > 0.05$). (**F**) Percentage of GP^{CL} bound to liposomes with varying PS concentration and fixed BMP and Ch. GP^{CL}-liposome binding was evaluated in the absence (grey; 1 mM EDTA) or presence of 1 mM Ca²⁺ (orange). (**G**) The ratio of bound GP^{CL} in the presence of Ca²⁺, to bound GP^{CL} in the absence of Ca²⁺ for liposomes of varying PS concentration (as in (F)). Data were fitted to a line with slope 0.003 (red line) with a 95% confidence interval of -0.002 to 0.008 (black dashed lines), which was not significantly different from zero ($p = 0.12$). (D-G) Data points or bars are presented as the mean ± standard error determined from three sets of measurements.

26.7 ± 0.2% in the absence of Ca²⁺. Stepwise increase in Ca²⁺ to 1 mM led to 52.0 ± 0.3% bound GP^{CL} (**Fig 1D**). However, the Ca²⁺-dependence of GP^{CL} binding to the PC:PS:BMP:Ch membrane was only apparent at acidic pH; Ca²⁺ did not promote GP^{CL}-membrane interaction at neutral pH (**Fig 1E**). These data demonstrate that the anionic phospholipids PS and BMP promote GP^{CL} binding to model membranes in a pH- and Ca²⁺-dependent manner.

Based on these results, we next hypothesized that BMP might be the primary driver of the $Ca^{2+}$-dependence of $GP^{CL}$ binding to the membrane. We therefore probed the relative contributions of PS and BMP to the observed $GP^{CL}$-membrane interaction. We titrated PS from 0 to 40mol%, while keeping the BMP and Ch concentrations constant, and assessed $GP^{CL}$ binding to liposomes in the absence and presence of 1 mM $Ca^{2+}$. Liposomes without PS bound low levels of $GP^{CL}$ irrespective of the presence of $Ca^{2+}$ (Fig 1F). Stepwise increases in PS concentration resulted in concomitant increases in $GP^{CL}$ binding to liposomes, always with greater binding in the presence of $Ca^{2+}$. Calculation of the ratio of bound $GP^{CL}$ in the presence of $Ca^{2+}$ to that seen in the absence of $Ca^{2+}$ across the PS titration indicated a linear trend with a slope not significantly different from zero ($p$ = 0.12; Fig 1G). These data demonstrate that PS promotes $GP^{CL}$-membrane binding, and thus enhances the robustness of our *in-vitro* measurements. But PS is not responsible for the $Ca^{2+}$ dependence. Instead, BMP is the more critical determinant of the $Ca^{2+}$-dependence of $GP^{CL}$-membrane interaction.

## Conformational dynamics of $GP^{CL}$ during membrane binding

We next sought to characterize the conformational changes of $GP^{CL}$ that enable interaction with the membrane under the conditions identified in our FCS assay. To this end, we used a previously established smFRET imaging assay that reports on conformational changes in GP2 (Fig 2A and 2B) [12]. We used lentiviral pseudoparticles with GP containing a non-natural amino acid, *trans*-cyclooct-2-ene-L-lysine (TCO*) at positions 501 and 610, which enabled labeling with Cy3- and Cy5-tetrazine [28]. We first assessed the conformational equilibrium of $GP^{CL}$ at pH 7.5 in the absence of $Ca^{2+}$. Advancements in single-molecule detection allowed us to resolve additional FRET states beyond what had previously been reported and improved our resolution in the estimation of state occupancies (Materials and Methods) [12]. Hidden Markov modelling (HMM) of the individual smFRET traces indicated the existence of four conformational states with FRET efficiencies of 0.24 ± 0.08, 0.49 ± 0.08, 0.72 ± 0.08, and 0.92 ± 0.08 (Fig 2C). The 0.92-FRET state (high FRET), which is consistent with the pre-fusion conformation reflected in structures of GP [2,29,30], predominated with 35 ± 2% occupancy (Fig 2D and 2E). HMM analysis indicated spontaneous transitions among the four FRET states and enabled construction of a transition density plot (TDP), which displays the relative frequency of transitions (Fig 2F). The TDP indicates that most transitions occur into and out of the 0.92-FRET state (high FRET), and between the 0.72- and 0.49-FRET states. Transitions into and out of the 0.24-FRET state (low FRET) were comparatively rare.

Given the sites of fluorophore attachment, we hypothesized that the low-FRET state might reflect a conformation in which the fusion loop was released from the hydrophobic cleft to positions where it can engage the target membrane. The 0.72- and 0.49-FRET states may reflect intermediate conformations where the fusion loop is destabilized prior to release from the hydrophobic cleft. Acidification to pH 5.5 reduced the occupancy of the high-FRET pre-fusion conformation to 14 ± 2% (Fig 2D and 2E). The occupancies in the 0.49- and low-FRET states increased and more frequent transitions occurred into and out of the low-FRET state. The addition of 1 mM $Ca^{2+}$ increased the kinetics of transitions into and out of the low-FRET state, while minimally affecting the thermodynamics of the FRET distribution (Fig 2D–2F).

Next, we incubated the labelled pseudovirions with PC:PS:BMP:Ch liposomes at pH 5.5 in the presence of 1 mM $Ca^{2+}$. Our FCS experiments had indicated robust membrane binding under these conditions. Interaction with liposomes decreased the occupancy in the high-FRET pre-fusion conformation, as well as the 0.72- and 0.49-FRET states, while increasing the low-FRET state occupancy to 54 ± 2% (Fig 2D and 2E). Overall dynamics of $GP^{CL}$ also decreased significantly with remaining transitions occurring in and out of the low-FRET state (Fig 2F).

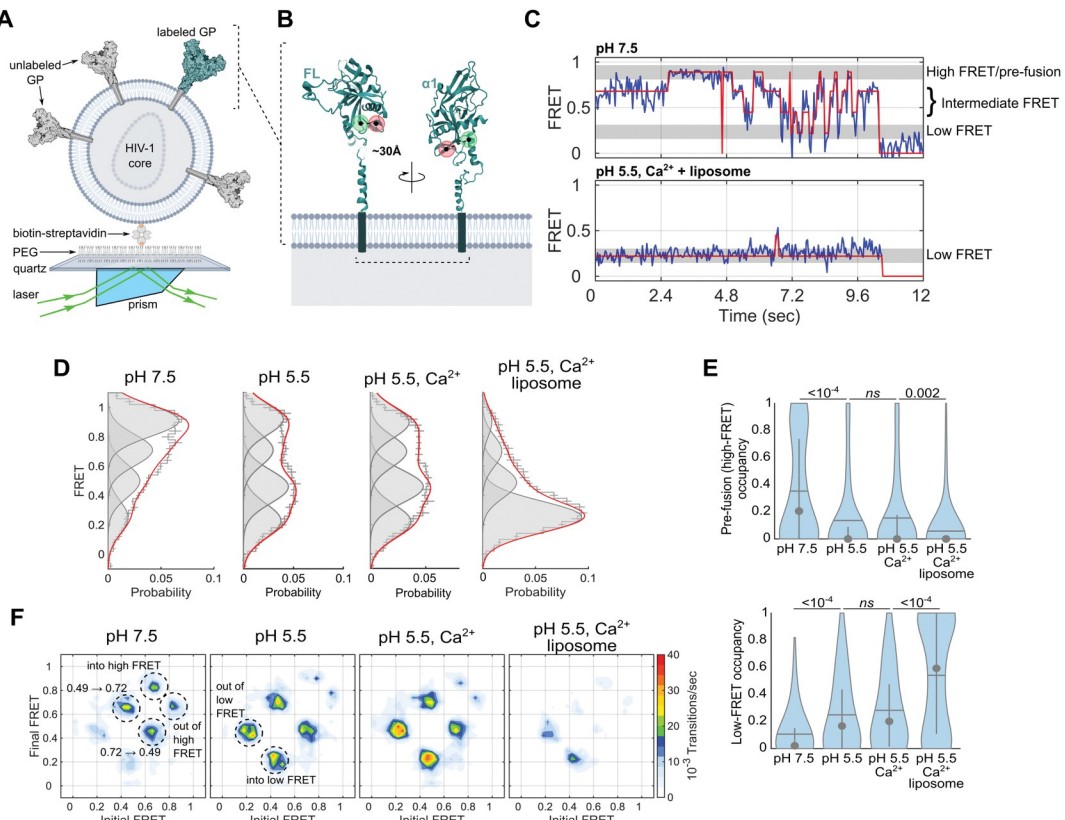

**Fig 2. Acidic pH, Ca²⁺ and target membrane control GP conformational dynamics.** (**A**) Experimental setup for smFRET imaging of pseudovirions labelled with Cy3 and Cy5 fluorophores. Labelled pseudovirions were immobilized on a passivated quartz microscope slide via a biotin-streptavidin linkage and imaged using prism-based TIRF microscopy. (**B**) Fluorophore attachment positions (red and green circles; residues 501 and 610 in GP2) are indicated on a GP protomer (PDB: 5JQ3). The indicated distance of 30 Å was previously determined through MD simulation of the fluorophore-labelled trimer [12]. (**C**) Representative FRET trajectories (blue) of GP^CL overlaid with idealization (red) determined through HMM analysis. The trajectory acquired at pH 7.5 (top) shows transitions between the pre-fusion conformation (0.92 ± 0.08 FRET), two intermediate-FRET states (0.72 ± 0.08 and 0.49 ± 0.08), and a low-FRET state (0.24 ± 0.08). In contrast, the FRET trajectory acquired in the presence of liposomes at pH 5.5 and 1 mM Ca²⁺ shows predominantly low FRET indicating a conformation stabilized by the presence of a target membrane. (**D**) Histograms of experimentally observed FRET values for wild-type GP^CL under the conditions indicated. Error bars indicate the mean probability per bin determined from three independent groups of traces. Overlaid on the histograms are single Gaussian distributions (grey, shaded) indicating the FRET states observed, which were generated during HMM analysis of the individual FRET trajectories. The sum of the four Gaussian distributions is also overlaid in red. The zero-FRET state, which arises due to photobleaching of the fluorophores is omitted to enable accurate visualization of the low-FRET state occupancy. (**E**) Violin plots indicating the distribution of occupancy in the high-FRET pre-fusion (top) and low-FRET (bottom) states. Overlaid on each plot is the median (circle), mean (horizontal line), and 25th and 75th quantile (vertical lines). Indicated $p$-values were determined by one-way ANOVA (ns, not significant). (**F**) TDPs displaying the relative frequency of transitions of GP^CL under the conditions indicated. Specific transitions between the observed FRET states are labelled with dashed circles.

Taken together, these data identify the low-FRET state as a GP^CL conformation that is enriched at pH 5.5 and stabilized by interaction with a target membrane, suggesting the release of the fusion loop from the hydrophobic cleft.

## MD simulations predict GP residues involved in sensing changes in pH

We next used molecular modelling to predict residues in the trimeric GP^CL ectodomain that sense changes in pH and control the conformational transitions necessary for membrane binding. We ran constant-pH molecular dynamics (MD) simulations using atomic coordinates of

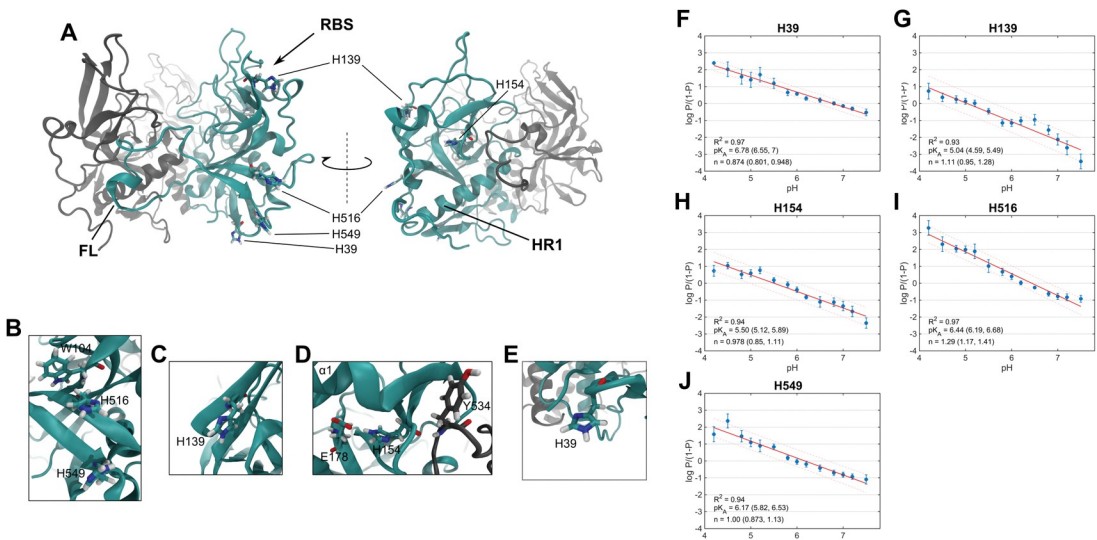

**Fig 3. Constant-pH MD simulations predict p$K_a$ values of histidines in GP$^{CL}$.** (**A**) Cartoon rendering of the trimeric GP$^{CL}$ ectodomain (PDB: 5HJ3). One protomer within the trimer is highlighted in cyan. Also indicated are structural landmarks (FL, fusion loop; RBS, receptor-binding site; HR1, heptad repeat 1) and the 5 histidine residues that were considered in constant-pH MD simulations. (**B-E**) Zoomed-in views of the 5 histidines and their surrounding residues. (**F**) Hill plot for H39 generated from the simulated titration of the pH from 4.2 to 7.8. The fractional protonation (P) at each pH point is presented as the average ± standard error determined from three simulation runs. The data were fitted to a linear model, resulting in estimation of the p$K_a$ and Hill coefficient (n), which are presented with 95% confidence intervals in parentheses. The fit line is shown in red with 95% confidence intervals (dashed lines). Hill plots and fits for (**G**) H139, (**H**) H154, (**I**) H516, and (**J**) H549 are displayed as in (**F**).

the trimeric GP$^{CL}$ ectodomain [30,31]. These simulations enabled estimation of the fractional protonation state of each protonatable residue in GP$^{CL}$ at pH values ranging from 4.2 to 7.5. The resulting simulated titrations were fit to the Hill equation to determine estimates of the p$K_A$s and Hill coefficients for each residue. Simulations of trimeric GP$^{CL}$ were run in triplicate, providing a total of 9 sets of p$K_A$s and Hill coefficients. Not surprisingly, our simulations indicated that the five histidines in the GP$^{CL}$ ectodomain—H39, H139, H154, H516, and H549— had p$K_A$ values of 6.78, 5.04, 5.50, 6.44, and 6.17, respectively (**Fig 3**). These data suggest that these five histidines become protonated, likely at different times, during trafficking of EBOV through the endocytic pathway.

We next used equilibrium MD simulations to generate specific hypotheses about how the histidines contribute to mediating pH-induced GP$^{CL}$ conformational changes. Following minimization and a multi-step equilibration procedure (Materials and Methods), simulations of GP$^{CL}$ were run in triplicate for 250 ns. Thus, we obtained 750 ns of aggregate sampling for each of the three protomers in the GP$^{CL}$ trimer with either deprotonated or protonated histidines. The simulation of protonated histidines was meant to approximate the conditions of the acidic late endosome. We first calculated the root-mean-square-fluctuation (RMSF) for each residue in GP$^{CL}$, which quantifies the amplitude of dynamics. We then averaged the RMSFs across the three protomers and the three replicas for the unprotonated and protonated simulations. The difference in average RMSF for the two simulations reports on whether each residue was more dynamic in the unprotonated or protonated simulation. The fusion loop residues D522 through E540, which includes the hydrophobic patch shown to interact directly with the membrane [16,32], displayed greater dynamics in the protonated simulation, suggesting destabilization of the fusion loop in the hydrophobic cleft (**Fig 4A–4C**). Residue Y534 is notable due to its proximity to H154 of the neighbouring protomer. The simulations predict an

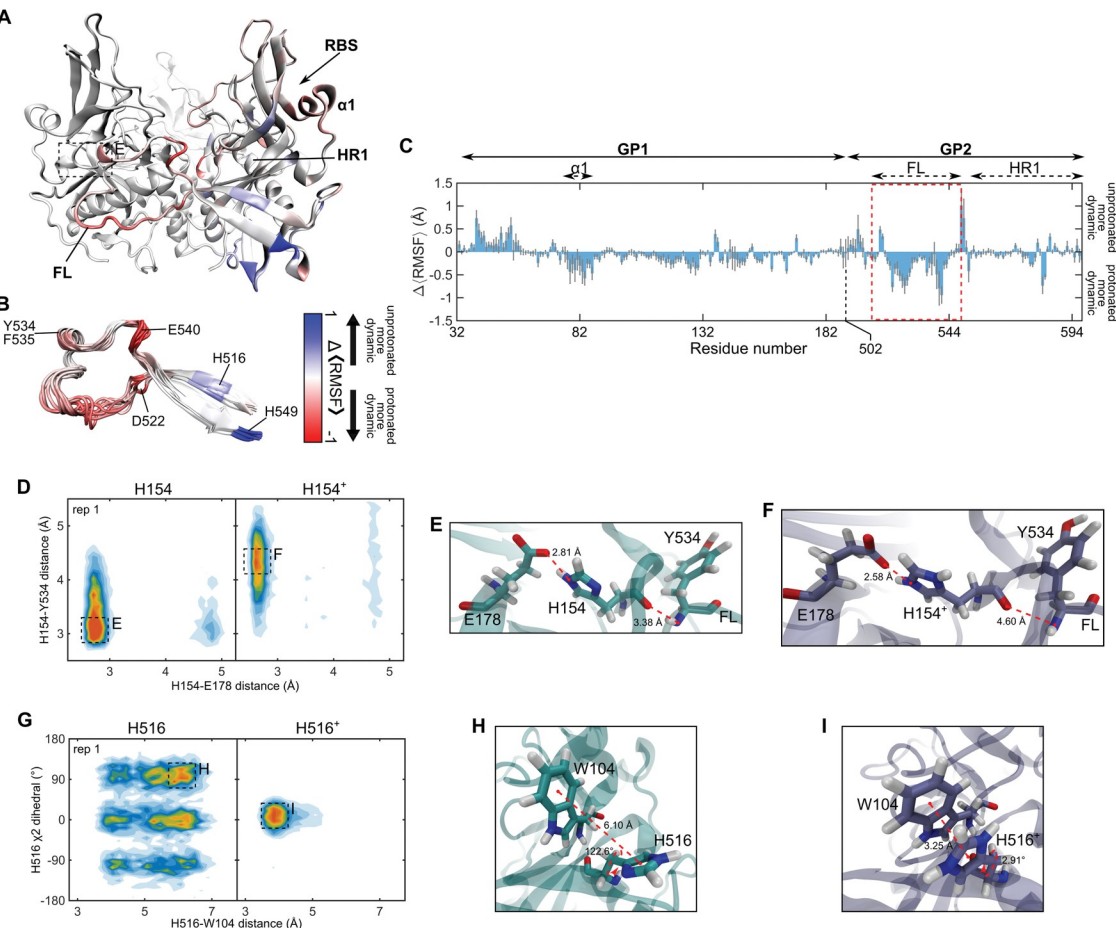

**Fig 4. MD simulations predict interactions that mediate GP^CL fusion loop stability.** (**A**) Cartoon rendering of the trimeric GP^CL ectodomain (PDB: 5HJ3). One protomer within the trimer is highlighted with a colored representation of the ΔRMSF. Red indicated regions that are more dynamic in the protonated simulation; blue indicates more dynamics in the deprotonated simulation. Also indicated are structural landmarks (FL, fusion loop; RBS, receptor-binding site; HR1, heptad repeat 1; α1 helix). (**B**) Overlay of nine fusion loops from the three simulations of GP^CL trimers colored according to the ΔRMSF. Key residues, which are discussed in the text, are highlighted. (**C**) ΔRMSF calculated from the simulations of GP^CL with protonated and deprotonated histidines. The ΔRMSF for the fusion (FL) is highlighted with a red dashed box. Data are presented as the mean ± standard error determined from the three replicate simulations. (**D**) Contour plot indicating the position of H154 in terms of the distance between the H154 and E178 sidechains, and between the H154 and Y534 backbones. Data are shown for one of the three replicas of the protonated (H154^+) and deprotonated (H154) simulations; data for the other two replicas are shown in **S1 Fig**. Predominant configurations are highlighted with dashed boxes and labelled with E and F. (**E**) Zoomed-in view of a simulation frame depicting the interactions between H154, E178, and Y534 in the configuration highlighted in (D), which predominated in the deprotonated simulation. (**F**) Corresponding zoomed-in view of the configuration highlighted in (D), which predominated in the protonated simulation. (**G**) Contour plot indicating the orientation of H516 in terms of the χ2 sidechain dihedral and the distance between the H516 and W104 side chains. Predominant conformations are highlighted with dashed boxes. Data for the other two replicas are shown in **S1 Fig**. (**H**) Zoomed-in view of a simulation frame depicting the interactions between H516 and W104 in the configuration highlighted in (G), which predominated in the deprotonated simulation. (**I**) Corresponding zoomed-in view of the configuration highlighted in (G), which predominated in the protonated simulation.

electrostatic interaction between the backbones of Y534 and H154, which may stabilize the fusion loop in the hydrophobic cleft. At the same time, the H154 side chain is engaged in electrostatics with the E178 side chain. Fluctuations in the H154-E178 distance decrease after protonation of H154, suggesting stabilization of the H154-E178 interaction. Additionally, the distance between H154 and Y534 increased upon protonation of H154 from approximately 3.4 to 4.6 Å (**Fig 4D–4F**). This analysis suggests the hypothesis that protonation of H154 in GP1

aids in release of the GP2 fusion loop of the neighbouring protomer by weakening the interaction with Y534.

We also noted that H516, which flanks the hydrophobic patch in the fusion loop, is engaged in a Pi-Pi stacking interaction with the side chain of W104 in GP1. The simulation suggests that this interaction is labile when H516 is deprotonated, with H516 sampling multiple conformations (**Fig 4G and 4H**). These dynamics can be visualized in terms of the distance between H516 and W104, and the $\chi_2$ dihedral angle of the H516 side chain. Protonation of H516 stabilized the stacking interaction with W104, reducing the dynamics of H516 and selecting a single pre-existing conformation (**Fig 4G and 4I**). These data suggest that GP1-GP2 interaction may be critical to GP2 conformational changes that remove the fusion loop from the hydrophobic cleft and stabilize it in a position competent for engagement with the membrane. While the remaining histidines—H39, H139, and H549—likely undergo changes in their protonation state during virus internalization and trafficking, our MD simulations do not indicate that protonation alters their dynamics or local interactions. We therefore consider it unlikely that these residues play a central role in mediating pH-dependent conformational changes in trimeric GP^CL. Going forward, we focused our attention on H154 and H516 and the predictions made by our simulations.

## H154 in GP1 mediates GP2 conformation at acidic pH

We next used our FCS and smFRET assays to test the predictions of our MD simulations regarding the role of H154 in mediating the conformational changes in trimeric GP^CL that enable membrane binding. First, we introduced an alanine substitution at residue H154, which eliminates the putative interaction with E178. Consistent with a previous study [33], full-length GP lacking the mucin domain (GPΔmuc) with the H154A mutation expressed at a lower level than wild-type GPΔmuc, resulting in reduced—though readily detectable—incorporation into pseudovirions (**S2 Fig**). Also consistently, the H154A mutant showed undetectable infectivity. Previous work also implicated H154A in poor proteolytic processing by cathepsin L [34]. Our use of the HRV3C protease likely alleviated this limitation. When the mutation was introduced in the context of our soluble GPΔTM ectodomain the native tertiary structure was verified by ELISA using KZ52, an antibody specific to the native tertiary structure of GP^CL (**S3 Fig**) [2,35]. In our FCS assay, the H154A mutant bound PC:PS:BMP:Ch liposomes at a low level at pH 7.5, similarly to wild-type GP^CL, irrespective of the presence of $Ca^{2+}$ (**Fig 5A**). smFRET imaging revealed a conformational landscape that included the same four FRET states as observed for wild-type GP^CL, further evidence that the mutation did not induce a folding defect (**Fig 5B**). However, the relative occupancies of the four FRET states were distinct from wild-type and dynamics were increased overall. The occupancy of the pre-fusion conformation was reduced from 35 ± 2% for wild-type to 25 ± 3% for H154A, while the occupancy in the 0.49-FRET state increased from 19 ± 2% for wild-type to 30 ± 2% for H154A (**Fig 5B and 5G**). Reducing the pH to 5.5 failed to promote any further membrane binding, again regardless of the presence of $Ca^{2+}$ (**Fig 5A**). While a slight increase in transitions to the low-FRET state was seen in the TDP, no effect on the FRET distribution could be detected (**Fig 5B, 5G and 5I**). These data suggest that the H154A mutation destabilizes the pre-fusion conformation but prevents pH-induced stabilization of the low-FRET state, consistent with its lack of binding to liposomes.

To further test the importance of the electrostatic interaction between H154 and E178, we introduced the positively charged H154K mutation. We hypothesized that the electrostatic interaction between H154K and E178 would stabilize the low-FRET conformation without dependence on pH. We observed only minimal interaction of the H154K mutant with the

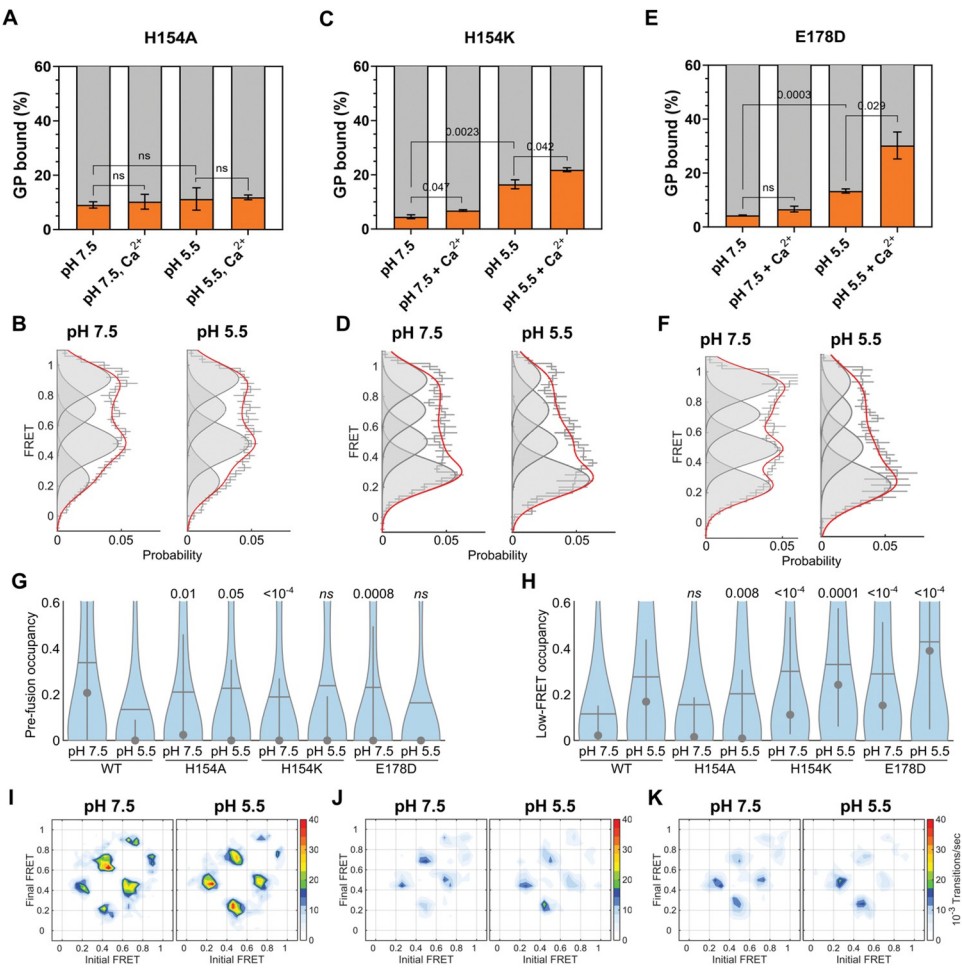

**Fig 5. The H154-E178 interaction controls release of the fusion peptide to enable membrane binding.** (**A**) Percentage of total GP^CL containing the H154A mutation bound to PC:PS:BMP:Ch liposomes under the indicated conditions determined by FCS. Indicated *p*-values were determined by *t*-test (ns, *p* > 0.05). (**B**) FRET histograms, displayed as in **Fig 2D**, for the H154A mutant at the indicated pH. (**C**) FCS and (**D**) smFRET data for the H154K mutant, indicating GP^CL membrane binding and conformation, respectively. (**E**) FCS and (**F**) smFRET data for the E178D mutant, indicating GP^CL membrane binding and conformation, respectively. (**G**) Violin plots indicating the distribution in high-FRET pre-fusion state occupancy and (**H**) low-FRET state occupancy, displayed as in **Fig 2E**. (**I**) TDPs indicating the frequency of FRET transitions at the indicated pH for the H154A mutant, the (**J**) H154K mutant, and the (**K**) E178D mutant.

membrane at neutral pH. Addition of Ca²⁺ induced a small, but statistically significant, increase in membrane binding. At acidic pH, we observed an increase in membrane binding, which was further enhanced by the presence of Ca²⁺ (**Fig 5C**). Again, our smFRET data indicated the same four FRET states as seen for wild-type GP^CL, confirming no loss of native folding. But the occupancy in the high-FRET pre-fusion conformation was approximately 50% lower than wild-type at neutral pH, and the low-FRET occupancy was approximately 300% greater than wild-type (**Fig 5D, 5G and 5H**). This significant shift in the conformational equilibrium is also reflected by the loss of recognition by KZ52. Given the location of the KZ52 epitope, this is consistent with displacement of the fusion loop and the HR1 helix [2]. Reduction in pH had minimal effect on the conformation of the H154K mutant, with only a slight increase in transitions to low FRET (**Fig 5J**). These data suggest that the presence of a positive

charge at position 154 is a key determinant of the stable formation of the low-FRET state. However, this shift in conformational equilibrium toward low FRET was not sufficient to enable membrane binding at neutral pH, suggesting that protonation of additional residues is still critical.

To further validate the interactions between H154, E178, and Y534 in controlling GP$^{CL}$ conformation, we introduced the E178D mutation. We hypothesized that the shorter side chain of the aspartate amino acid would draw H154 away from Y534 and reduce the stability of the fusion loop. FCS experiments demonstrated similar enhancement of membrane binding by acidic pH and Ca$^{2+}$ as seen for wild-type (**Fig 5E**). As expected, smFRET data showed reduced occupancy in the high-FRET pre-fusion conformation, at neutral pH as compared to wild-type. Moreover, reduction in pH increased occupancy in the low-FRET state to an extent greater than seen for wild-type (**Fig 5F and 5G**). Reduced transitions into high FRET further demonstrated the relative instability of the pre-fusion conformation (**Fig 5K**). Consistent with our prediction, these results suggest that upon protonation of H154, its interaction with E178D strengthens, which pulls H154 away from Y534 to an extent greater than wild-type. This may lead to premature release of the fusion loop at acidic pH, which could be reflected by low infectivity (**S2C Fig**). Taken together, these results support the hypothesis suggested by our MD simulations that H154 is a pH sensor that contributes to the regulation of GP conformational changes through its interactions with E178 and Y534.

## Protonation of H516 in GP2 further controls GP conformation

We next tested the role of H516-W104 interaction in mediating GP$^{CL}$ conformation and membrane binding. The H516A mutation, which should weaken the interaction with W104, showed only low levels of membrane binding in our FCS assay, with only a slight but statistically significant increase upon acidification ($p = 0.03$), and no Ca$^{2+}$ dependence (**Fig 6A**). In contrast, our smFRET data demonstrated no significant difference in the occupancies of the high-FRET pre-fusion conformation or the low-FRET state as compared to wild-type at either of the pHs tested (**Fig 6B and 6I**). This suggests that the H516A-W104 interaction is sufficient to stabilize the low-FRET conformation, but insufficient to support membrane binding. In contrast, the H516R mutation, which introduces a constitutive positive charge at position 516 that enables electrostatic interaction with W104, showed robust pH-mediated membrane binding, comparable to wild-type ($p > 0.05$; **Fig 6C**). Similarly, the H516W mutation, which can form a hydrophobic stacking interaction with W104, also demonstrated enhancement of membrane binding at pH 5.5 (**Fig 6E**). None of the H516 mutations demonstrated Ca$^{2+}$ dependence during membrane binding. In our smFRET experiments, at neutral pH both H516R and H516W showed reduced occupancy in the high-FRET pre-fusion conformation, and increased occupancy in the low-FRET state, as compared to wild-type. This is consistent with the interaction between residues 516 and 104 contributing to GP conformational changes without dependence on acidic pH. For both H516R and H516W mutants, acidification promoted the low-FRET state to an extent greater than for wild-type (**Fig 6D, 6F, 6I and 6J**), perhaps indicating the combined effect of protonation of H154 and stabilization of the interaction between residues 516 and 104.

To further interrogate the H516-W104 interaction, we introduced the W104E mutation, which we predicted would support an electrostatic interaction with H516 and would be stabilized at acidic pH. The W104E mutant showed only a slight, but statistically significant ($p = 0.037$), enhancement in membrane binding at pH 5.5, which may indicate suboptimal positioning of the fusion loop (**Fig 6G**). Consistent with our prediction, the W104E mutation showed greater occupancy in low FRET than wild-type at both pHs tested, with the anticipated

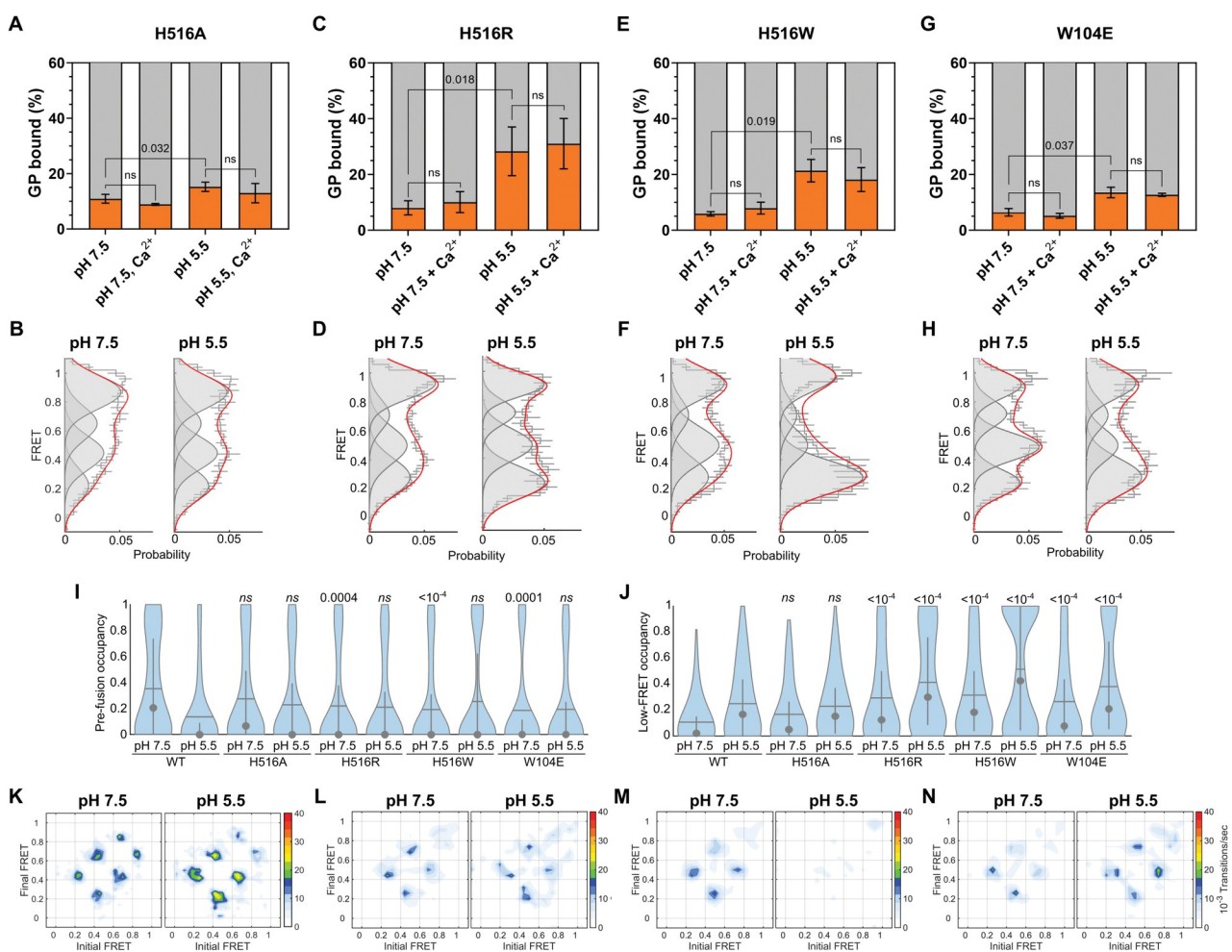

**Fig 6. The H516-W104 interaction stabilizes the fusion peptide position for membrane binding.** (**A**) Percentage of total GP$^{CL}$ containing the H516A mutation bound to PC:PS:BMP:Ch liposomes under the indicated conditions determined by FCS. Indicated *p*-values were determined by *t*-test (ns, *p* > 0.05). (**B**) FRET histograms, displayed as in **Fig 2D**, for the H516A mutant at the indicated pH. (**C**) FCS and (**D**) smFRET data for the H516R mutant, indicating GP$^{CL}$ membrane binding and conformation, respectively. (**E**) FCS and (**F**) smFRET data for the H516W mutant, indicating GP$^{CL}$ membrane binding and conformation, respectively. (**G**) FCS and (**H**) smFRET data for the W104E mutant, indicating GP$^{CL}$ membrane binding and conformation, respectively. (**I**) Violin plots indicating the distribution in high-FRET pre-fusion state occupancy and (**J**) low-FRET state occupancy, displayed as in **Fig 2E**. (**K**) TDPs indicating the frequency of FRET transitions at the indicated pH for the H516A mutant, the (**L**) H516R mutant, the (**M**) H516W mutant, and the (**N**) W104E mutant.

increase at pH 5.5 (**Fig 6H and 6J**). Taken together, these data support the hypothesis that an interaction between H516 and W104 is critical for pH-mediated GP$^{CL}$ conformational changes. Stabilization of this interaction can contribute to efficiently forming the conformation reflected by the low-FRET state. Mutations to H516 and W104 led to reduction in infectivity, likely indicating that subsequent conformational changes necessary for fusion are inhibited (**S2C Fig**). Finally, these data suggest that formation of the low-FRET conformation is not sufficient to support membrane binding, as evidenced by the conformational equilibrium of H516A being indistinguishable from wild-type. This implicates the local conformation of the fusion loop, which is not visualized in the current assays, in also contributing to mediating interaction with phospholipids. A non-native fusion loop conformation may explain the lack of Ca$^{2+}$ dependence seen for the H516 mutants.

## Acidic residues in the fusion loop are critical to GP$^{CL}$ interaction with the membrane

In the data presented above, mutations to residue H516 resulted in $Ca^{2+}$ no longer promoting interaction with the membrane even though the low-FRET state was stabilized under acidic conditions. These results suggest that $Ca^{2+}$ does not influence the global conformation of GP but rather may have a localized effect on the conformation of GP2 or the fusion loop. Indeed, $Ca^{2+}$ had minimal effect on the conformation of wild-type GP$^{CL}$ at pH 5.5 as determined by our smFRET assay (**Fig 2D and 2E**). Previous work identified conserved acidic residues (D522, E523, E540 and E545) that mediated interaction of the fusion loop with $Ca^{2+}$ [14]. We therefore sought to test the role of these residues in the context of trimeric GP$^{CL}$ through introduction of alanine substitutions. As above, native GP$^{CL}$ tertiary structure was evaluated using KZ52 ELISA (**S3 Fig**). E523A and E545A were unstable in solution and were not considered further.

The D522A mutant bound to liposomes to a comparable extent as wild-type GP$^{CL}$ at neutral pH (**Fig 7A**). smFRET analysis indicated a lower occupancy in the high-FRET pre-fusion conformation as compared to wild-type (**Fig 7B and 7E**). Slightly increased dynamics as compared to wild-type were apparent in the TDP, suggesting that the mutation destabilizes GP2 overall (**Fig 7G**). While membrane binding was increased slightly at pH 5.5, it failed to reach the level of wild-type (**Fig 7A**). The conformational equilibrium and dynamics were similar to wild-type at acidic pH, suggesting that the D522A mutation had no effect on the sensitivity to pH (**Fig 7B and 7E**). As expected, the addition of $Ca^{2+}$ had no effect on membrane binding, conformation, or dynamics of D522A. The E540A mutation performed similarly to wild-type and D522A in membrane binding at pH 7.5. However, in contrast to D522A, E540A showed robust membrane binding comparable to wild-type at pH 5.5. Interestingly, $Ca^{2+}$ inhibited E540A membrane binding at pH 5.5 (**Fig 7C**). This suggests that E540A interacts with $Ca^{2+}$ but fails to adopt a fusion loop conformation that is competent for membrane insertion. Our smFRET data indicated a conformational equilibrium like D522A, in which the high-FRET pre-fusion conformation was destabilized as compared to wild-type at pH 7.5, but no differences in FRET state occupancies were detected at acidic pH irrespective of $Ca^{2+}$ (**Fig 7D–7H**). These results indicate that the putative $Ca^{2+}$-binding site mediates the stability of the pre-fusion GP$^{CL}$ conformation at neutral pH but does not contribute to GP$^{CL}$ conformational changes upon acidification. The observation that the stability of the low-FRET state for the D522A and E540A mutants is comparable at pH 5.5, while their membrane-binding capacities diverge is further indication that transition to the low-FRET state is insufficient to support membrane binding. The importance of $Ca^{2+}$ coordination likely comes after the low-FRET conformation is achieved and prior to interaction with the target membrane.

## Discussion

Here, we report that EBOV senses chemical cues in the late endosome that promote GP$^{CL}$ conformational changes needed to initiate membrane fusion. We find that anionic lipids that are enriched in the late endosome facilitate GP$^{CL}$-membrane binding, with BMP being especially important in mediating the $Ca^{2+}$-dependence. We used smFRET imaging to visualize the pH-induced conformational changes in the GP2 subunit that enable interaction with the membrane. Molecular simulations informed hypotheses, which were confirmed experimentally, suggested that H154 in GP1 and H516 in GP2 respond to changes in pH and promote conformational changes in the GP$^{CL}$ trimer. Given the differences in estimated pKas, protonation of H154 and H516 may occur in series and control the stepwise promotion of conformational changes. A precise structural description of the low-FRET conformation is not available. Yet,

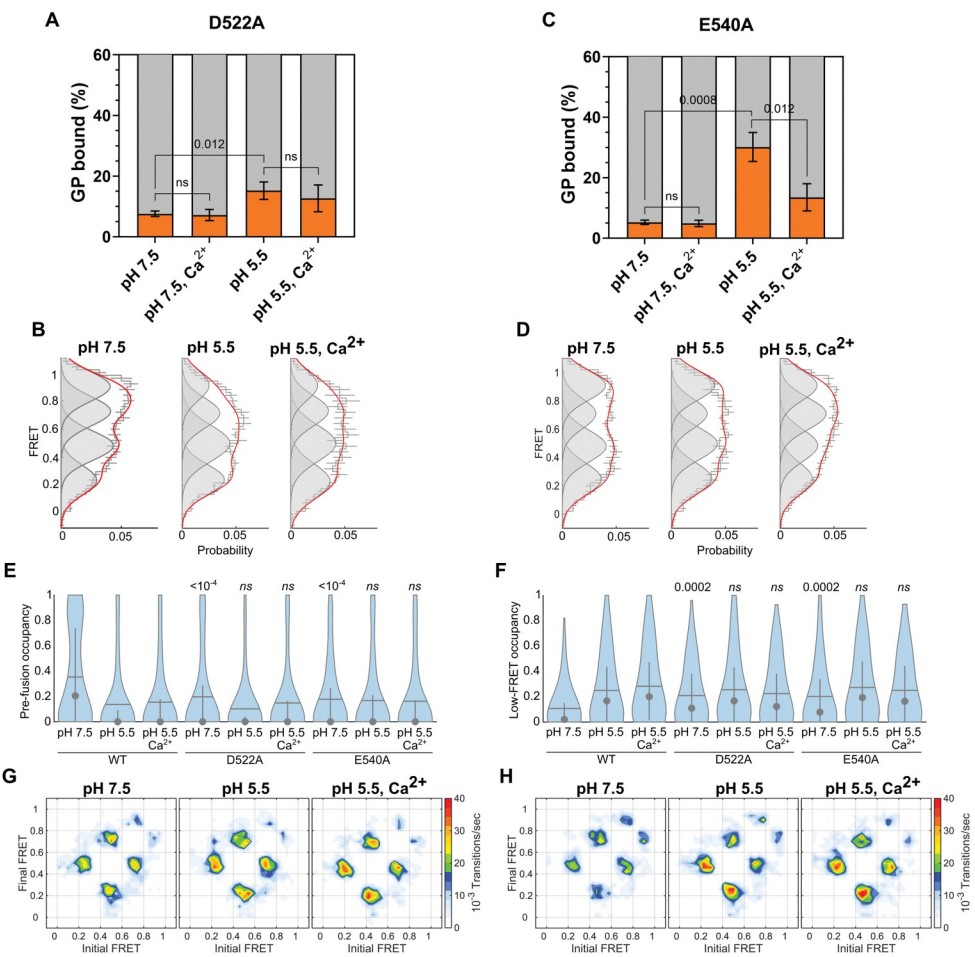

**Fig 7. D522 and E540 control Ca²⁺-mediated GP^CL membrane binding.** (**A**) Percentage of total GP^CL containing the D522A mutation bound to PC:PS:BMP:Ch liposomes under the indicated conditions determined by FCS. Indicated *p*-values were determined by *t*-test (ns, $p > 0.05$). (**B**) FRET histograms, displayed as in **Fig 2D**, for the D522A mutant at the indicated conditions. (**C**) FCS and (**D**) smFRET data for the E540A mutant, indicating GP^CL membrane binding and conformation, respectively. (**E**) Violin plots indicating the distribution in high-FRET pre-fusion state occupancy and (**F**) low-FRET state occupancy, displayed as in **Fig 2E**. (**G**) TDPs indicating the frequency of FRET transitions at the indicated conditions for the D522A mutant, and the (**H**) E540A mutant.

our results indicate that the extent of GP^CL-membrane binding is inversely correlated (Pearson correlation $r = -0.54$, $p = 0.025$) with the occupancy of GP^CL in the pre-fusion conformation (**Fig 8**), which underscores the relevance of the observed conformational changes to the fusion reaction.

We found that GP^CL -membrane interaction was promoted by PS and BMP. The unique structure of BMP alters the negative spontaneous curvature of membranes [36]. Additionally, BMP increases the net surface charge of liposomes, enhancing their interaction with Ca²⁺ [37]. Both features of BMP may promote efficient interaction between GP^CL and the target membrane. We speculate that GP^CL could preferentially interact with BMP via Ca²⁺ coordination within the fusion loop. Since BMP is exclusively present in the late endosome [37], it could also regulate the timing of fusion of EBOV by avoiding pre-mature engagement of the fusion loop with a membrane. Dependence on BMP in regulating the location and timing of fusion has been reported for other enveloped viruses such as Dengue virus, Lassa virus, vesicular stomatitis virus, and influenza virus [20,21,23,38]. Whether downstream events during EBOV

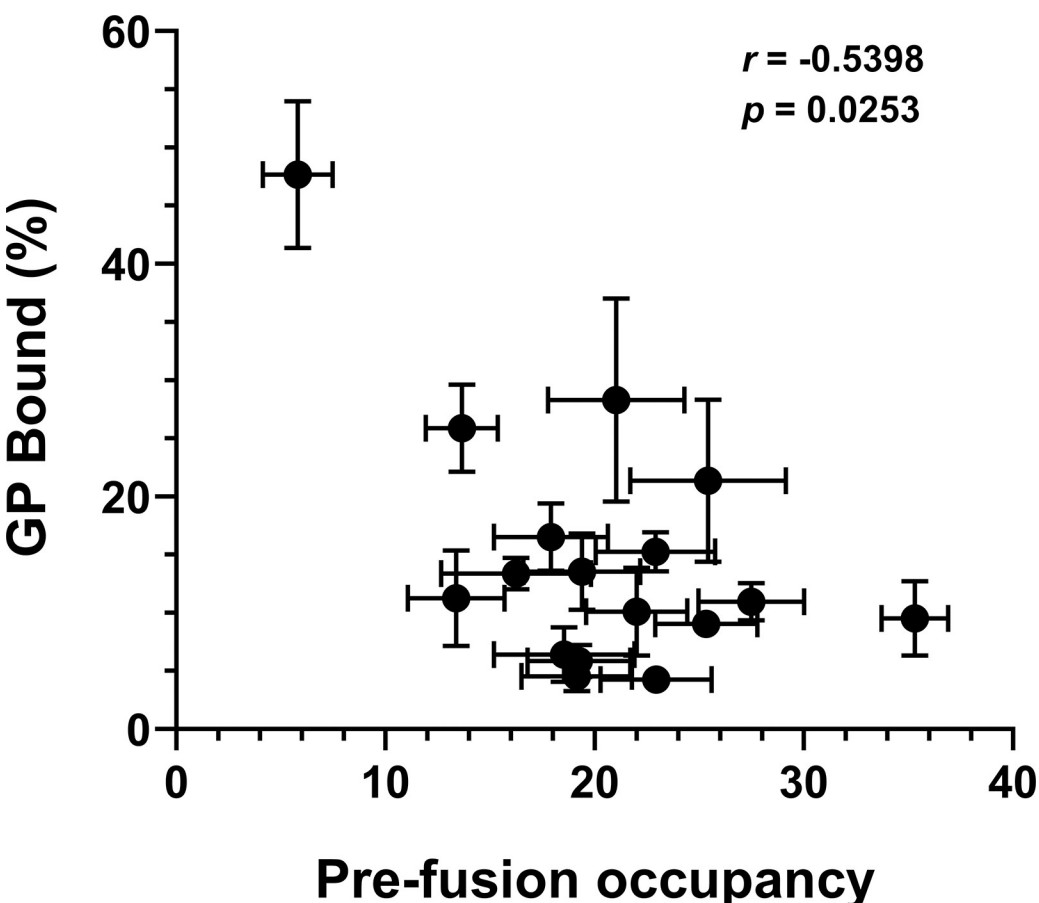

**Fig 8. GP<sup>CL</sup> occupancy in the pre-fusion high-FRET conformation is inversely correlated with the extent of membrane binding.** High-FRET pre-fusion state occupancy was determined through HMM analysis of smFRET trajectories. Membrane binding was determined by FCS. The Pearson's correlation coefficient (*r*) and associated *p*-value are indicated. Data for D522A and E540A (grey points) does not correlate with membrane binding and were excluded from the analysis.

fusion are also accelerated by BMP, as reported for Lassa virus, will be the topic of future studies [21]. PS is also known to interact with $Ca^{2+}$, which can lead to changes in membrane curvature [39]. This altered curvature could create a membrane that is not conducive to $GP^{CL}$ binding, which may explain the reduction in $GP^{CL}$ interaction with PC:PS:Ch liposomes at elevated $Ca^{2+}$ concentrations (**Fig 4A**) [40,41]. We utilized an elevated PS concentration to optimize the extent of $GP^{CL}$-membrane interaction *in vitro* and increase the dynamic range of our measurements. Additional components of the endosomal membrane, which are yet to be identified, might further promote interaction with $GP^{CL}$ such that elevated PS would not be necessary. Nonetheless, our data indicate that the elevated PS does not impact the pH- or $Ca^{2+}$-dependence of the $GP^{CL}$-membrane interaction.

A previous study evaluated the role of BMP during GP-mediated cell-cell fusion, concluding that BMP had no effect [9]. The authors sought to incorporate BMP into the plasma membrane in the presence of delipidated BSA. This approach does not enable precise control over the molar percentage of BMP, in part because delipidated BSA can transport lipid moieties into the cell lumen, which may limit incorporation into the plasma membrane [37,42]. Additionally, the overall lipid environment of the plasma membrane is distinct from the endosome. Lipids such as PS, which our experiments suggest promotes the GP-membrane interaction,

and PE are enriched in the inner leaflet of the plasma membrane. In contrast, they are distributed in both inner and outer leaflets in endosomes [19,43]. Hence, in a cell-cell fusion assay, any role played by such lipids in GP-mediated fusion could be obscured. Our FCS-based assay enabled precise control over the concentration of the lipids employed. Nonetheless, while our experiments indicate that BMP promotes GP-membrane interactions, subsequent steps in the fusion reaction may require unidentified factors that are not present in the plasma membrane, which might have masked the effect of BMP in previous cell-cell fusion assays.

Protonation of histidine residues triggers membrane binding and fusion of influenza virus, Semiliki Forest virus, and human metapneumoviruses [44,45]. Here, we report that residues H154 and H516 tune the responsiveness of EBOV GP$^{CL}$ to acidic pH through both stabilizing and destabilizing mechanisms. Histidine residues can interact electrostatically with cationic amino acids such as lysines or arginines, forming His-Cat pairs, which are known to stabilize the pre-fusion conformations of several viral envelope glycoproteins [45]. Protonation of histidines destabilizes the His-Cat interaction, which contributes to triggering conformational changes necessary for viral fusion. In contrast, H154 and H516 are engaged in interactions not described by the His-Cat paradigm. We show here that protonation of H154 stabilizes its interaction with the E178 side chain, which pulls H154 away from Y534 in the fusion loop of the neighboring protomer. Destabilization of the backbone electrostatics between H154 and Y534 contributes to the release of the fusion loop from the hydrophobic cleft, which facilitates the fusion loop-membrane interaction.

A previous study concluded that H516 had no direct role in the fusion loop-membrane interaction and that a positive charge at this position was insufficient to induce fusion loop-mediated membrane fusion [16]. Consistent with this study, we observed similar membrane binding for the H516R mutant as for wild-type GP, but with approximately 50% loss of infectivity. In this way, our results support the previous study that protonation of H516 alone is not sufficient to promote robust fusion. Instead, our results suggest that H516 mediates trimeric GP$^{CL}$ conformation under acidic conditions. The apparent inter-domain Pi-Pi stacking interaction between H516 and W104, which is stabilized by protonation of H516, may aid in ensuring proper positioning of the fusion loop for initial interaction with the membrane. Formation of an electrostatic or hydrophobic interaction of W104 with the H516R and H516W mutants, respectively, supported adoption of the low-FRET conformation and membrane binding. Thus, we propose that protonation of H516 aids in making the fusion loop accessible to the membrane in the context of trimeric GP$^{CL}$. The putative H516-W104 interaction implies that GP1 remains associated with GP2 until after engagement with the target membrane. NPC1 binding to GP1 may partially serve to localize EBOV on the endosomal membrane to increase the efficiency with which the fusion loop inserts into the membrane. Given previous studies indicating that acidic pH does not trigger GP-mediated fusion [9,46], we speculate that events downstream of the GP-membrane interaction may not be pH dependent. A recent study suggested the proteolytic activity of cathepsins during the final stages of fusion [15].

$Ca^{2+}$ can assist in membrane fusion through coordination by anionic residues in the fusion loop, as seen for Rubella virus, MERS-CoV, SARS-CoV-1 and -2, and EBOV [14,47–50]. This may stabilize a fusion loop conformation that is optimal for insertion into the membrane. In addition, $Ca^{2+}$ coordination in the fusion loop can affect the physical properties of the target membrane in a manner that promotes fusion [49]. Previous studies have shown that elevated $Ca^{2+}$ can inhibit EBOV entry and GP function by an unknown mechanism [12,51]. While estimates of the $Ca^{2+}$ concentration in the late endosome vary considerably, previous cell-based and *in-vitro* studies of the EBOV fusion loop or the intact GP trimer have utilized concentrations in the range of 0.5–2 mM [9,12,14], which may exceed the physiological concentration [52–54]. To ensure results that are comparable with the literature, we opted to maintain this

precedence in our experiments. The overall consistency in our results from measurements of membrane binding and infectivity suggest that alternative $Ca^{2+}$ concentrations would not affect the interpretation of our data. Moreover, the titration of $Ca^{2+}$ indicated that BMP promotes $GP^{CL}$-membrane binding across the physiological range of concentrations.

The importance of anionic residues, D522 and E540, in mediating the $Ca^{2+}$-dependence of fusion loop-membrane interaction and lipid mixing was previously reported [14]. Our findings support these conclusions and confirm that residues D522 and E540 are critical mediators of the $Ca^{2+}$-dependence of trimeric $GP^{CL}$-membrane interaction. The results presented here provide the additional insight that these residues contribute to stabilization of the pre-fusion conformation, but their coordination of $Ca^{2+}$ does not drive large-scale $GP^{CL}$ conformational changes. Rather, we propose that acidic pH is the primary driver of $GP^{CL}$ conformational changes that are necessary for interaction with the membrane, whereas $Ca^{2+}$ binding likely has a localized effect on the fusion loop. Structural studies of the intact trimer, performed under conditions that promote membrane fusion, are needed to elucidate the GP2 structure, the relative position of the GP1 domain, and the local fusion loop configuration that enables engagement with the target membrane. Biophysical interrogations such as those presented here, likely involving smFRET imaging with alternative fluorophore attachment sites, should guide these future studies by identifying conditions that promote functional conformations and the residues that mediate their stability.

## Material and methods

### Cell lines

Expi293F cells (Gibco, ThermoFisher Scientific, Waltham, MA) were cultured in Expi293 expression medium in an orbital shaking incubator at 37°C, 8% $CO_2$, 125rpm. HEK293T FirB cells, which have high furin expression, were a kind gift from Dr. Theodore C. Pierson (Emerging Respiratory Virus section, Laboratory of Infectious Diseases, NIH, Bethesda, MD) [55]. These cells were cultured in DMEM (Gibco, ThermoFisher Scientific, Waltham, MA) with 10% cosmic calf serum (Hyclone, Cytiva Life Sciences, Marlborough, MA) and 1% penicillin-streptomycin (Gibco, ThermoFisher Scientific, Waltham, MA) at 37°C, 5% $CO_2$.

### Plasmids

pHLsec-GPΔTM and pMAM51-GPΔmuc plasmids were obtained from Dr. Kartik Chandran's lab (Einstein College of Medicine, NY). pHLsec-GPΔTM encodes EBOV (Mayinga) GP sequence (UniProt Q05320) with deleted mucin-like and transmembrane domains. A T4 fibritin foldon trimerization domain and 6X-histidine tag for Ni-NTA purification has been inserted into the C terminus. The A1 (*GDSLDMLEWSLM*) and A4 (*DSLSMLEW*) peptides were introduced at positions 32 and 501 in GP1 and GP2, respectively, for site-specific labelling of GPΔTM, as previously described [26]. pMAM51-GPΔmuc encodes full-length GP with the mucin-like domain deleted and was used for all pseudovirion experiments. In both GPΔTM and GPΔmuc the thermolysin cleavage site, *VNAT* at position 203, was replaced with an HRV3C protease recognition site (*LEVLFQGP*) by site directed mutagenesis (Q5 site directed mutagenesis kit, New England Biolabs, Ipswich, MA). All amino acid substitutions were also introduced in GPΔTM and GPΔmuc via site-directed mutagenesis. pNL4.3.Luc.R-E- used in infectivity assays was obtained through the NIH AIDS Reagent program (contributed by Dr. Nathaniel Landau, New York University School of Medicine) [56]. An amber stop codon (TAG) in the *tat* gene was modified by site-directed mutagenesis to –TAA to prevent readthrough during incorporation of TCO* for labelling of GPΔmuc. Plasmids PyIRS^AF and

eRF1 were provided by Dr. Edward Lemke (Johannes Gutenberg-University of Mainz, Germany).

## Protein expression and purification

For production of GPΔTM proteins (wild-type and mutants), Expi293F cells were transfected with pHLsec-GPΔTM using polyethyleneimine (PEI MAX, Polysciences, Warrington, PA) at a mass ratio of 1:3 DNA:PEI MAX. As previously described, a 2:1 ratio of pHLsec-GPΔTM to tagged pHLsec-GPΔTM was transfected to ensure that GPΔTM trimers contained on average a single tagged protomer. The supernatants containing soluble GPΔTM proteins were harvested 5 days post-transfection. The proteins were purified using Ni-NTA agarose beads (Pierce, ThermoFisher Scientific, Waltham, MA). The protein was bound to the column in phosphate-buffered saline (PBS) containing 10mM imidazole, followed by washing with 20mM imidazole in PBS and elution in 200 mM imidazole containing PBS. Following purification, proteins were exchanged to labelling buffer (20 mM HEPES, 50 mM NaCl, pH 7.5) using VivaSpin 6 concentrator (Sartorius AG, Gottingen, Germany).

## Labelling of GPΔTM

Wild-type and mutant GPΔTM proteins were labelled with 5 μM LD650 conjugated to coenzyme A (LD650-CoA; Lumidyne Technologies, New York, NY). The fluorophore was attached to the A1 and A4 peptides in the tagged GPΔTM through incubation with 5 μM acyl carrier protein synthase (AcpS) in the labelling buffer with 10 mM $Mg(CH_3COO)_2$ overnight at room temperature [26]. The labelled proteins were subjected to overnight cleavage by exogenous furin (New England Biolabs, Ipswich, MA) at 37˚C to fully convert GP0 to GP1 and GP2. The processed proteins were purified by size-exclusion chromatography on a Superdex 200 Increase 10/300 GL column (GE Healthcare, Chicago, IL). Purified, labelled proteins were concentrated using Amicon Ultra 30K filters (MilliporeSigma, Burlington, MA), aliquoted and stored at -80˚C until further use. The concentration of proteins was determined using Bradford reagent.

## Indirect ELISA

Labelled proteins were diluted to a final concentration of 5 μg/ml in PBS and coated onto the wells of a 96-well polystyrene plate (Pierce, ThermoFisher Scientific, Waltham, MA) by incubating overnight at 4˚C. The plate was washed three times with PBST (PBS with 0.1% Tween-20) followed by blocking with 5% skim milk in PBST for 3h at room temperature. The blocking solution was removed, and the plate was again washed twice with PBST. Proteins were probed with KZ52 antibody [26,35] at a dilution of 1:1000 overnight at 4˚C. The plate was then washed and incubated with horseradish peroxidase-conjugated anti-human IgG (Invitrogen, ThermoFisher Scientific, Waltham, MA) at a dilution of 1:2000 for 2h at room temperature. After washing the plate four times with PBST, TMB solution (3,3',5,5'-tetramethylbenzidine; ThermoFisher Scientific, Waltham, MA) was added to each well, incubated for 15 min, followed by addition of an equal volume of 2M sulphuric acid. The optical density was immediately read at 450 nm in a Synergy H1 microplate reader (BioTek, Winooski, VT).

## Liposome preparation

The following lipids were used for liposome preparation: POPC (1-palmitoyl-2-oleoyl-glycero-3-phosphocholine), POPS (1-palmitoyl-2-oleoyl-sn-glycero-3-phospho-L-serine (sodium salt)), BMP (bis(monooleoylglycero)phosphate (S,R Isomer)) and cholesterol (all from Avanti

Polar lipids, Alabaster, AL). Stock solution of lipids (10 mg/ml) were diluted at desired ratios in chloroform to obtain a final total lipid concentration of 1 mM. A lipid film was formed in a glass vial by evaporating the chloroform under a steady stream of Argon gas. Residual chloroform was removed by incubating the lipid film overnight under vacuum. Lipids were rehydrated with 5 mM HEPES, 10 mM MES and 150 mM NaCl, pH 7.5, for 1 hour at room temperature. The lipid suspension was vortexed 5–7 times in 10 sec pulses followed by 10 freeze-thaw cycles with liquid nitrogen. Liposomes were formed by extruding the lipid solution 37 times through a 100 nm polycarbonate membrane (Whatman Nucleopore track-etched membrane) in a mini-extruder (Avanti Polar Lipids, Alabaster, AL). The size of liposomes was verified using dynamic light scattering (Zetasizer Nano, Malvern Panalytical, Malvern, UK). The liposomes were stored at 4˚C and used within a week.

### HRV3C cleavage

Thermolysin is commonly used to remove the glycan cap from GP1 in place of the endosomal cathepsin proteases [57]. However, this led to removal of the fluorophore from labelled GPΔTM. To alleviate this off-target effect of thermolysin, the thermolysin cleavage site was replaced with the HRV3C protease recognition sequence. The glycan cap was thus removed from GPΔTM and pseudovirions with GPΔmuc through incubation with HRV3C protease (Pierce, ThermoFisher Scientific, Waltham, MA) at 10˚C for 16 hr, which left the fluorescent labelling intact (S2A Fig). Cleavage of LD650-labelled GPΔTM was verified by in-gel fluorescence imaging of uncleaved and cleaved protein on a 4–20% polyacrylamide using a Chemi-Doc MP imaging system (Bio-Rad, Hercules, CA) followed by Coomassie staining. In a pseudovirion infectivity assay, introduction of the HRV3C sequence left GP 85% functional as compared to wild-type GP (S2B Fig).

### Fluorescence correlation spectroscopy

For membrane binding studies, liposomes (500 μM total lipid) were incubated with 5 nM labelled GP$^{CL}$in 5 mM HEPES, 10 mM MES, 150 mM NaCl (pH 7.5 or 5.5). For experiments performed in the absence of Ca$^{2+}$, 1mM EDTA was included to chelate any Ca$^{2+}$ already bound to GP$^{CL}$. Otherwise, the indicated concentration of CaCl$_2$ (0-1mM) was included in the incubation. The liposome-GP$^{CL}$ mixture was incubated at 37˚C for 20 min to allow binding. For experiments performed at pH 5.5, the liposome mixture was acidified with 1 M HCl prior to addition of GP$^{CL}$. FCS experiments were performed by dropping 50 μL of the liposome-GP$^{CL}$ mixture onto a coverslip (No. 1.5 Thorlabs, Newton, NJ). To prevent sticking of the protein-membrane complex to the glass surface, coverslips were plasma-cleaned followed by coating with 10% polyethylene glycol (PEG-8000, Promega, Madison, WI). 100 autocorrelation curves, 5 sec each in length, were recorded at room temperature using a 638 nm laser in a Cor-Tector SX100 (LightEdge Technologies Ltd., Zhongshan City, China). The curves were fitted to the following model for two species diffusing in three dimensions with triplet blinking [25, 58, 59]:

$$G(\tau) = G_{triplet}(\tau).G_{complex}(\tau), \text{ where}$$

$$G_{triplet}(\tau) = 1 - T + T \cdot e^{\frac{\tau}{\tau_{triplet}}}, \text{ and}$$

$$G_{complex}(\tau) = \frac{1}{N}\left( f \cdot \left[1 + \frac{\tau}{\tau_{protein}}\right]^{-1}\left[1 + \frac{\tau}{s^2\tau_{protein}}\right]^{-1/2} + \alpha \cdot (1 - f) \cdot \left[1 + \frac{\tau}{\tau_{liposome}}\right]^{-1}\left[1 + \frac{\tau}{s^2\tau_{liposome}}\right]^{-1/2}\right)$$

where, $N$ is the number of molecules in the confocal volume, $f$ is the fraction of free protein, $\alpha$ is the average brightness of the protein-liposome complex, $\tau_{protein}$ is the diffusion time of free protein, $\tau_{liposome}$ is the diffusion time of the liposome, and $s$ is the structural parameter, which reflects the dimensions of the confocal volume. The diffusion times, $\tau_{protein}$ and $\tau_{liposome}$ were calculated by fitting the autocorrelation curves of GPΔTM and liposomes (labelled with 40nM DiD) separately to single species diffusion models. The values of $s$ and $\tau_{protein}$ were kept constant to calculate the fraction of free and bound protein for all samples. Due to polydispersity of liposomes and fast photophysical dynamics of triplet blinking, $\tau_{liposome}$ and $\tau_{triplet}$ were allowed to vary during fitting. The analysis was carried out in MATLAB (MathWorks, Natick, MA) using a non-linear least-square curve fitting algorithm. All values were averaged over three independent experiments. Two-tailed, independent Student's T-tests were performed to calculate the statistical significance between mutations under the conditions tested ($\alpha$ = 0.05).

## Western blotting

All proteins and pseudovirions were run on 4–20% denaturing polyacrylamide gels (Bio-Rad, Hercules, CA) and transferred to nitrocellulose membrane using Trans-blot Turbo (Bio-Rad, Hercules, CA). Membranes were rinsed with PBST, blocked with 5% skim milk in PBST for 1 h followed by probing of GP and p24 with monoclonal antibody (mAb) H3C8 at a dilution of 1:1000, [60] and mouse mAB B1217M at a dilution of 1:2000 (Genetex, Irvine, CA), respectively. The mAb H3C8 was humanized by cloning its variable heavy and light chain fragments in human IgG expression vectors obtained from Dr. Michel Nussenzwieg (The Rockefeller University). Membranes were washed three times with PBST, incubated with horseradish peroxidase-conjugated anti-human IgG (Invitrogen, ThermoFisher, Waltham, MA) and anti-mouse IgG (ThermoFisher, Waltham, MA) for 1 h at room temperature and developed with SuperSignal West Pico PLUS chemiluminescent substrate (ThermoFisher Scientific, Waltham, MA).

## Infectivity assay

Infectivity of VSV pseudovirions containing wild-type GPΔmuc was compared to pseudovirions containing GPΔmuc[HRV3C] via flow cytometry [61]. HEK293T cells were transfected with GPΔmuc and GPΔmuc[HRV3C] plasmid using polyethyleneimine (PEI MAX, Polysciences, Warrington, PA) at a mass ratio of 1:3 PEI MAX:DNA. Cells were transduced with VSVΔG-GFP-VSVG pseudovirions 24 h after transfection. Supernatants containing VSVΔG-GFP-GPΔmuc and VSVΔG-GFP-GPΔmuc[HRV3C] were collected 24 h post-transduction and filtered through 0.45 μm filter. As a negative control, bald particles were generated by transducing cells not expressing GPΔmuc. Vero cells were infected with VSV pseudovirions through incubation at 37˚C for 1 h with gentle agitation every 15min to allow even spread of pseudovirions. Fresh media was added, and infection was allowed to proceed for five more hours. Cells were trypsinized and assayed for expression of GFP using a flow cytometer (MACSQuant Analyzer 1.0). The results were averaged across technical replicates and standard error was calculated from biological replicates.

To assess infectivity of GPΔmuc[HRV3C] mutants, pseudovirions with luciferase-expressing HIV-1 core were produced. Plasmids pMAM51-GPΔmuc[HRV3C] and pNL4.3.Luc.R-E- were co-transfected at a ratio of 1:5 in HEK 293T FirB cells. Pseudovirions were harvested 48 h post-transfection by collecting the supernatant, passing through a 0.45 μm filter, and layering on 10% sucrose in PBS solution followed by ultracentrifugation at 25000 rpm for 2 h at 4˚C. After resuspension of the pellet in PBS, particles were analyzed by western blot. The virions were cleaved with HRV3C to remove the glycan cap as above, incubated with Vero cells for 5 h

at 37°C, followed by replacement of growth media. After 48 h, cells were lysed with Glo Lysis Buffer (Promega, Madison, WI) for 5 min at room temperature. Luciferase activity was recorded by mixing equal volumes of cell lysate and Steady Glo Reagent (Promega, Madison, WI) and reading in a Synergy H1 plate reader (Biotek, Winooski, VT). The luminescence was normalized to expression of GP for each sample. Bald pseudovirions containing only pNL4.3. Luc.R-E- were used as a negative control.

## Pseudovirion production and fluorescent labelling

To facilitate attachment of fluorophores to GP on the surface of pseudovirions for smFRET imaging, the non-natural amino acid TCO* (SiChem GmbH, Bremen, Germany) was introduced at positions 501 and 610 through amber stop codon suppression (GP*) (**Fig 2B**) [28]. Pseudovirions were produced by transfecting HEK293T FirB cells with pMAM51-GPΔ-muc$^{HRV3C}$ plasmids with and without amber stop codons at a 1:1 ratio, which equated to an excess of GP protein over GP*. This ratio was optimized to ensure that the pseudovirions rarely contained more than a single GP* protomer per particle. A plasmid encoding HIV-1 GagPol was also transfected to provide the pseudovirion core. To increase the efficiency of amber codon readthrough, plasmids eRF1 and PyIRS$^{AF}$ were also included in transfection [12,28]. The supernatant containing pseudovirions was harvested 48 h post-transfection, filtered through a 0.45 μm mixed cellulose ester membrane and layered onto 10% sucrose (in PBS) solution. The pseudovirions were pelleted by ultracentrifugation at 25000 rpm for 2 h at 4°C. Pseudovirions were resuspended in 500 μL PBS and incubated with 500 nM Cy3- and Cy5-tetrazine (Jena Biosciences, Jena, Germany) for 30 min at room temperature. 60 μM DSPE-PEG2000 biotin (Avanti Polar Lipids, Alabaster, AL) was added to the labelling reaction and incubated for another 30 min at room temperature with gentle mixing. The labelling reaction was layered on a 6–30% OptiPrep (Sigma-Aldrich, MilliporeSigma, Burlington, MA) density gradient and ultracentrifuged at 35000 rpm for 1 h at 4°C. Labelled pseudovirions were collected, aliquoted, and analyzed by western blot.

## smFRET imaging assay and data analysis

All smFRET experiments were performed following removal of the glycan cap from GP. Labelled and glycan cap-cleaved pseudovirions were immobilized on streptavidin-coated quartz slides and imaged on a wide-field prism-based TIRF microscope [62]. Imaging was performed in the same buffer used for the membrane binding assay (5 mM HEPES, 10 mM MES, 150 mM NaCl [pH 7.5 or 5.5]). To study the effect of Ca$^{2+}$, the buffer was supplemented with 1 mM CaCl$_2$. smFRET data was acquired at room temperature at 25 frames/sec using the Micro-Manager microscope control software (micromanager.org). Analysis of smFRET data was performed using the SPARTAN software package (https://www.scottcblanchardlab.com/software) [63] in Matlab (MathWorks, Natick, MA). smFRET trajectories were selected according to the following criteria: acceptor fluorescence intensity greater than 35; FRET was detectable for at least 15 frames prior to photobleaching; correlation coefficient between donor and acceptor fluorescence traces was less than -0.4; signal-to-noise ratio was greater than 10; and background fluorescence was less than 50. Trajectories that met these criteria were further verified manually and fitted to a 5-state linear hidden Markov model (including a zero-FRET state) using maximum point likelihood (MPL) algorithm implemented in SPARTAN [64]. The 5-state linear model was chosen based on the Akaike Information Criterium (AIC) and the Bayesian Inference Criterium (BIC) (**S4 Fig**) [65,66]. Several models were initially considered with varying numbers of model parameters and topology. The 5-state linear model minimized both the AIC and BIC criteria relative to the models considered and was thus chosen for

analysis. The idealizations from total number of traces for each sample were used to calculate the occupancies in different FRET states and construct the FRET histograms and transition density plots (TDPs).

## MD simulations

A model of trimeric GP$^{CL}$ was generated from atomic coordinates determined through x-ray crystallography (PDB accession: 5HJ3) [30]. The model included residues 32–188 of GP1 and 502–598 of GP2. The protein components of the models were parameterized with the CHARMM36m forcefield using CHARMM-GUI. The systems were charge neutralized and solvated with the TIP3P explicit solvent model. The solvated proteins were energy minimized for 0.1 ns, followed by equilibration using a stepwise protocol [67]. Briefly, the protein backbone was harmonically constrained, with the constraints being released stepwise over 4 0.3-ns intervals. Equilibration was run in the NPT ensemble with temperature maintained at 310 K and pressure maintained at 1 atm using the Langevin thermostat and the Nose-Hoover Langevin barostat, respectively. Electrostatics were calculated using the Particle Mesh Ewald algorithm. Constant-pH MD simulations were run using the *cph* module in NAMD v2.14 on the SCI cluster at UMass Chan Medical School [31]. Titrations were simulated in triplicate with 14 pH points ranging from 4.2 to 7.5. Protonation switches were attempted every 1 ps with switch times of 20 ps for 1000 cycles, yielding a total simulation time of 21 ns for each of three replicas. Inherent pKa values were assigned with PropKa. The results of the constant-pH simulations were analyzed using the *cphanalyze* program in the *pynamd* analysis package. Apparent p$K_a$s and Hill coefficients were determined by linear fitting in Matlab. Equilibrium MD simulations were run for 250 ns in the NPT ensemble using a GPU-accelerated installation of NAMD v3.0b2. on the SCI cluster at UMass Chan Medical School. Trajectories were visualized in VMD and analyzed using the Tcl scripting interface.

## Supporting information

**S1 Fig. Replicate data from MD simulation.** (**A**) Zoomed-in view of (purple) protonated and (cyan) deprotonated simulation frames depicting an overlay of the interactions between H154, E178, and Y534. (**B**) As in **Fig 4D**, contour plot indicates the position of H154 in terms of the distance between the H154 and E178 sidechains, and between the H154 and Y534 backbones. Data are shown for replicas two and three of the protonated (H154$^+$) and deprotonated (H154) simulations. (**C**) Zoomed-in view of (purple) protonated and (cyan) deprotonated simulation frames depicting an overlay of the interactions between H516 and W104. (**D**) As in **Fig 4G**, contour plot indicating the orientation of H516 in terms of the $\chi_2$ sidechain dihedral and the distance between the H516 and W104 side chains. Data are shown for replicas two and three of the protonated (H154$^+$) and deprotonated (H154) simulations.
(TIF)

**S2 Fig. Proteolysis, labelling, and infectivity of GP variants.** (**A**) Denaturing polyacrylamide gel showing uncleaved GPΔTM and GP$^{CL}$ (cleaved with HRV3C) imaged under Cy5 fluorescence (top) and stained with Coomassie (bottom). (**B**) Comparison of relative infectivity of GPΔmuc, and GPΔmuc with the HRV3C cleavage site. (**C**) (top) Western blot of pseudovirions showing expression of GPΔmuc (with the HRV3C cleavage site) and p24. (bottom) Relative infectivity of lentiviral pseudoparticles containing the mutant GPΔmuc. All infectivity experiments were performed with three biological replicates, each measured in triplicate. Data points represent the mean infectivity determined across biological replicates.
(TIF)

**S3 Fig. Antigenicity of mutant GPs.** (**A**) ELISA of wild-type and mutant GPΔTM proteins with mAb KZ52. Except for H154K, introduction of mutations in GPΔTM does not have a significant impact on the antigenicity of GP. ELISAs were carried out in triplicate. Data points represent absorbance values at 450 nm. Bars and error bars represent the mean and standard deviations.
(TIF)

**S4 Fig. Model selection through minimization of the AIC and BIC.** smFRET trajectories from wild-type GP$^{CL}$ were fit to a series of different models by maximum likelihood optimization using the MPL algorithm (see Materials and Methods). The maximized likelihood estimated for each model was corrected for differing numbers of model parameters using the AIC and BIC criteria. Under this procedure, the model with best fitness among those considered will generate minimum AIC and BIC values. Both criteria identified the 5-state linear model as providing the best representation of the data among the models considered. Overlaid on the plot are schematic representations of the kinetics models considered. In all models, the 0 state corresponds to the 0-FRET state, with the others reflecting non-zero FRET states. Lines represent connections (allowed transitions) between states. All non-zero FRET states are connected to the 0-FRET state since photobleaching is seen from all FRET states. TDPs shown in **Figs 2** and **5**–**7** support the identification of a linear model.
(TIF)

## Acknowledgments

The authors would like to thank Dr. Gang Han (UMass Chan Medical School, Worcester, MA) for assistance with dynamic light scattering.

## Author Contributions

**Conceptualization:** Aastha Jain, Ramesh Govindan, James B. Munro.

**Data curation:** Aastha Jain, James B. Munro.

**Formal analysis:** Aastha Jain, Alex R. Berkman, James B. Munro.

**Funding acquisition:** Jeremy Luban, James B. Munro.

**Investigation:** Aastha Jain, Ramesh Govindan, Alex R. Berkman, James B. Munro.

**Methodology:** Aastha Jain, Ramesh Govindan, James B. Munro.

**Resources:** Marco A. Díaz-Salinas.

**Software:** James B. Munro.

**Supervision:** James B. Munro.

**Visualization:** Aastha Jain, James B. Munro.

**Writing – original draft:** Aastha Jain, James B. Munro.

**Writing – review & editing:** Aastha Jain, Jeremy Luban, Marco A. Díaz-Salinas, Natasha D. Durham, James B. Munro.

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
