## [Decision Letter · Decision Letter 0]

20 Nov 2023

Dear Dr. Munro,

We are pleased to inform you that your manuscript 'Regulation of Ebola GP conformation and membrane binding by the chemical environment of the late endosome' has been provisionally accepted for publication in PLOS Pathogens.

Best regards,

Thomas Hoenen

Academic Editor

PLOS Pathogens

Meike Dittmann

Section Editor

PLOS Pathogens

Kasturi Haldar

Editor-in-Chief

PLOS Pathogens

orcid.org/0000-0001-5065-158X

Michael Malim

Editor-in-Chief

PLOS Pathogens

orcid.org/0000-0002-7699-2064

This revised version of the manuscript has been reviewed by two of the reviewers of the original version, both of which indicated that their concerns have been addressed. Responses to the remaining reviewer of the original manuscript version, who unfortunately was not available for a re-review, were assessed at the editorial level, and also found satisfactory.

Reviewer Comments (if any, and for reference):

Reviewer's Responses to Questions

**Part I - Summary**

Reviewer #1: The authors have well addressed the concerns I raised in the first version of the manuscript. Thank you, and I think this will be a valuable addition to the scientific literature.

Reviewer #2: (No Response)

**Part II – Major Issues: Key Experiments Required for Acceptance**

Reviewer #1: (No Response)

Reviewer #2: (No Response)

**Part III – Minor Issues: Editorial and Data Presentation Modifications**

Reviewer #1: (No Response)

Reviewer #2: (No Response)

PLOS authors have the option to publish the peer review history of their article (what does this mean?). If published, this will include your full peer review and any attached files.

Reviewer #1: No

Reviewer #2: No

---

## [Editor Report · Acceptance letter]

1 Dec 2023

Dear Dr. Munro,

We are delighted to inform you that your manuscript, "Regulation of Ebola GP conformation and membrane binding by the chemical environment of the late endosome," has been formally accepted for publication in PLOS Pathogens.

Best regards,

Michael Malim

Editor-in-Chief

PLOS Pathogens

orcid.org/0000-0002-7699-2064